# Investigation of Stability and Power Consumption of an AlGaN/GaN Heterostructure Hydrogen Gas Sensor Using Different Bias Conditions

**DOI:** 10.3390/s19245549

**Published:** 2019-12-16

**Authors:** June-Heang Choi, Hyungtak Kim, Hyuk-Kee Sung, Ho-Young Cha

**Affiliations:** School of Electronic and Electrical Engineering, Hongik University, Seoul 04066, Korea; petrus0314@mail.hongik.ac.kr (J.-H.C.); hkim@hongik.ac.kr (H.K.); hksung@hongik.ac.kr (H.-K.S.)

**Keywords:** AlGaN/GaN heterostructure, hydrogen sensor, power consumption, stability, current mode operation

## Abstract

A Pd-functionalized hydrogen gas sensor was fabricated on an AlGaN/GaN-on-Si heterostructure platform. The AlGaN layer under the Pd catalyst area was partially recessed by plasma etching, which resulted in a low standby current level enhancing the sensor response. Sensor stability and power consumption depending on operation conditions were carefully investigated using two different bias modes: constant voltage bias mode and constant current bias mode. From the stability point of view, high voltage operation is better than low voltage operation for the constant voltage mode of operation, whereas low current operation is preferred over high current operation for the constant current mode of operation. That is, stable operation with lower standby power consumption can be achieved with the constant current bias operation. The fabricated AlGaN/GaN-on-Si hydrogen sensor exhibited excellent sensing characteristics; a response of 120% with a response time of < 0.4 s at a bias current density of 1 mA/mm at 200 °C. The standby power consumption was only 0.54 W/cm^2^ for a sensing catalyst area of 100 × 24 μm^2^.

## 1. Introduction

Recently, hydrogen has attracted much attention as a new energy source, and extensive research has been conducted [1] into its use. Hydrogen is used as the material for fuel cells and has been highlighted as a pollution-free natural energy source [2]. However, great care must be taken with regard to the usage and handling of hydrogen because it is easily oxidized and explodes even at low concentrations of the order of ~4% [3]. Therefore, reliable hydrogen gas sensors with fast response characteristics are required to identify and prevent hydrogen leakage in fuel cells.

Semiconductor-based gas sensors have the benefits of low cost, small size, and low power consumption. Various semiconductor materials have been utilized to implement hydrogen sensors [4,5,6,7,8,9,10], among which GaN is an attractive material for a hydrogen sensor operating at high temperatures because of its wide energy bandgap with low intrinsic carrier density [11,12,13]. The low intrinsic carrier density allows GaN to maintain its semiconductor properties in high-temperature environments. Therefore, GaN is suitable for use as a gas sensor platform in cases where the gas–catalyst reaction processes require high temperatures. AlGaN/GaN heterostructure materials have the added benefit of high electron concentrations of a two-dimensional electron gas channel with a high channel mobility of >1500 cm^2^/V·s, which enhance the response speeds.

In this study, AlGaN/GaN heterostructure-based hydrogen gas sensors were fabricated using a Pd catalyst. Further, we investigated the sensing characteristics using both voltage and current bias mode operations.

## 2. Experiments

### 2.1. Sensor Fabrication

Field-effect transistor (FET)-type hydrogen gas sensors were fabricated on an AlGaN/GaN-on-Si heterostructure substrate. The epitaxial structure consisted of a 10 nm in situ SiN_x_ passivation layer, a 3.5 nm GaN cap layer, a 23 nm Al_0.24_Ga_0.76_N barrier layer, a 420 nm i-GaN layer, and a 4.2 μm GaN buffer layer on a 1000 μm thick (111)-oriented silicon substrate. The sensor fabrication process is schematically illustrated in Figure 1. After solvent cleaning, the ohmic contact areas were defined by photolithography, and the exposed in-situ SiN_x_ layer and underlying epitaxial layers were etched down to the middle of the AlGaN barrier layer using CF_4_ and Cl_2_/BCl_2_-based inductively-coupled plasma-reactive ion etching (ICP-RIE), respectively, prior to ohmic metal deposition. After Ti/Al/Ni/Au (= 200/1200/250/500 Å) metallization, rapid thermal annealing was carried out at 830 °C for 30 s in N_2_ ambient to achieve low ohmic contact resistance. The measured contact resistance and sheet resistance were 0.56 Ω⋅mm and 117 Ω/sq, respectively. After an additional photolithography process, a Ti/Au (= 20/250 nm) metal stack was evaporated for the pad electrodes. MESA isolation with an etch depth of 350 nm was conducted using the same etching process described above. After defining the catalyst area, the exposed in-situ SiN_x_ layer and underlying epitaxial layers were also etched using the same process. The remaining AlGaN barrier layer thickness under the catalyst area was 6 nm. The thinned AlGaN barrier layer has the advantage of low standby current level that can enhance the response of the device. A 30 nm thick Pd film was evaporated for the catalyst material; Pd is an excellent catalyst for hydrogen because of its high solubility of hydrogen [14]. A 100 nm SiN_x_ film was deposited at 190 °C using the plasma-enhanced chemical vapor deposition technique for surface passivation. The catalyst window area (= 24 × 100 µm^2^) for gas reaction was defined by a CF_4_ based SiN_x_ etching process, during which the pad contact areas were also opened. The fabricated sensor structure and its microscopic image are shown in Figure 2a,b, respectively.

### 2.2. Sensor Characterization 

Pd has high solubility to hydrogen and has phase transition characteristics with hydrogen atoms [15]. When hydrogen gas is introduced to the Pd catalyst, the hydrogen molecules are adsorbed on Pd and decomposed into atoms that are rapidly dissolved in Pd. As a result, α-PdH and β-PdH can be produced by the following reaction mechanism where more β-phase can be produced with increasing temperature [16].
(1)$$\left[\left(\mathrm{Pd}+H_{2}\underset{\;a\;}{\to }\alpha \text{-PdH}\right)+H_{2}\underset{\;b\;}{\to }\beta \text{-PdH}\right].$$

The β-phase has a larger lattice constant that increases hydrogen solubility [17]. As a result, hydrogen permeability increases with increasing temperature [18]. The dipole effects of the hydrogen atoms absorbed in Pd lower the surface potential of the AlGaN barrier layer, as illustrated in Figure 2c, which increases the two-dimensional electron gas concentration at the AlGaN–GaN interface. Consequently, the sensor current level increases with hydrogen absorption. This reaction is reversible and can recover the initial state when the hydrogen supply is terminated.

The fabricated sensor was placed in a gas chamber whose temperature was controlled by a heater module located below the chamber plate. Hydrogen gas at a concentration of 4% was used for sensor characterization, and the chamber temperature was varied from room temperature to 250 °C.

## 3. Results and Discussion

First, we investigated the current–voltage characteristics of the sensor, where the sensor was biased by a voltage supply and the current between the two electrodes was measured. Figure 3a–e shows the current–voltage characteristics at different temperatures with and without hydrogen injection. The measurements were repeated five times to assess reproducibility. The current level was observed to increase with hydrogen injection regardless of the temperature, which could be associated with the reduced surface potential due to the Pd–H chemical reaction. It should be noted that the current level changed slightly at room temperature and at 100 °C as the measurements were repeated, whereas no variations were observed at temperatures of 150 °C and higher.

The sensing response under the constant voltage bias operation can be defined by [19]
(2)$$\mathrm{Response}\;\left[\%\right]=\left(\frac{I_{gas}-I_{air}}{I_{air}}\;\right)\times 100,$$
where *I_gas_* is the current measured with hydrogen injection and *I_air_* is the standby current without hydrogen injection. The bias-voltage-dependent response characteristics are shown in Figure 3f as a function of the temperature. The response increased with temperature up to 200 °C, beyond which a relatively smaller increase was observed. The increase in response at elevated temperatures was associated with the enhanced dissociation of hydrogen molecules [18] The lower standby current at elevated temperatures was attributed to the lower channel mobility caused by photon scattering. It should also be noted that the response current level also decreased at elevated temperatures. Therefore, there is a trade-off for the sensor response between hydrogen permeability and mobility reduction with increasing temperature. The slower increase in response beyond 200 °C was associated with significant reduction in mobility [20]. A maximum response of 180% was achieved at 250 °C with a bias voltage of 1 V. However, it is suggested that the optimum reaction temperature is 200 °C, considering the power consumption of the heater module that has to be built into the sensor system.

Since the response depends strongly on the bias voltage, it is better to set the bias voltage in the stable region rather than at the maximum point because the sensor signal might drift over time. For comparison, we chose two bias conditions of the constant voltage bias operation for the stability test: The first was at 1 V with maximum response and the second was at 9 V in the saturation region. In order to investigate the stability of the sensor signal, sensing repeatability tests were carried out as follows. Hydrogen gas was introduced for 5 s, paused for 55 s, and the process was repeated for 30 min. Figure 4a,b shows the repeatability characteristics measured at the bias voltages of 1 V and 9 V, respectively. Although the response was high at the low bias voltage condition (i.e., 1 V), the sensing environment conditions, including the bias voltage fluctuation, caused noticeable differences. On the other hand, the response at the high voltage bias condition (i.e., 9 V) was less dependent of the bias voltage fluctuation, which is expected to result in more stable operation. The variation ranges in the peak sensing and bottom recovery current levels with a bias voltage of 1 V were 0.4 and 1.0 mA/mm, respectively, whereas those for the bias voltage of 9 V were 0.2 and 0.1 mA/mm, respectively. It should be noted that the most significant drawback of the high voltage bias operation is the relatively higher standby power consumption owing to the higher standby current and voltage levels. The standby power consumption for 1 and 9 V operations were 3.4 and 97.2 mW/mm, respectively, where the area was defined by the catalyst area. The response transient characteristics are shown in the magnified plots located on the right-hand side. The response times for both bias conditions were less than 0.4 s.

The sensitivity characteristics were investigated with various hydrogen concentration using the stable bias condition of 9 V. Figure 5a shows the sensor response versus hydrogen concentration. The hydrogen concentration was varied from 1000 to 40,000 ppm. The sensitivity can be defined by the sensor output response as a function of the measurand applied to the input [21]. We used the change in the sensor current level as the output response and the hydrogen concentration as the input measurand. As plotted in Figure 5b, a linear relationship between the response current change and the hydrogen concentration was observed up to 6000 ppm, beyond which the increase in the response current became smaller. The sensitivity was 1 µA/mm⋅ppm, which was derived from the linear regime.

We also investigated the current-dependent response characteristics, wherein the sensor was biased by a current supply and the voltage between the two electrodes was measured. Figure 6a–e shows the voltage–current characteristics at different temperatures with and without hydrogen injection. The measurements were repeated five times. The voltage levels decreased with hydrogen injection. Similar to the constant voltage operation case, the current–voltage characteristics at lower temperatures were relatively unstable and those at 200 and 250 °C were stable.

The sensing response under the constant bias current mode operation can be defined by [22]
(3)$$\mathrm{Response}\;=\left(\frac{V_{air}-V_{gas}}{V_{gas}}\;\right)\times 100,\;$$
where *V_air_* is the standby voltage without hydrogen injection and *V_gas_* is the voltage measured with hydrogen injection. The current-dependent response characteristics are plotted in Figure 6f as a function of temperature. The response increased significantly with temperature up to 200 °C and a relatively lower increase was observed at 250 °C.

For comparison, we chose two bias conditions of the constant current bias operation for the stability test at 200 °C: The first at 1 mA/mm in the stable region and the second at 11 mA/mm in the high-response region. Similar to the constant voltage mode tests, the sensing repeatability tests were carried out with hydrogen injection of 5 s and a period of 60 s for 30 min. Figure 7a,b shows the repeatability characteristics measured at bias currents of 1 and 11 mA/mm, respectively. Although the response was much higher for the bias current of 11 mA/mm, significant fluctuations were observed over the duration of the test. On the other hand, very stable operation was observed at 1 mA/mm. The variation ranges for the peak recovery and bottom sensing voltage levels with the bias current of 1 mA/mm were 1.21 and 1.12 mV, respectively, whereas those with the bias current of 11 mA/mm were 33 and 709 mV, respectively. The standby power consumption for the 1 and 11 mA/mm operations were 0.15 and 85 mW/mm, respectively. Based on our experiments, it is suggested that the constant current operation with low bias current level is a good choice for both stable and low power consumption operation.

The sensitivity characteristics were investigated with various hydrogen concentration using the stable bias condition of 1 mA/mm. Figure 8a shows the sensor response versus hydrogen concentration. The hydrogen concentration was varied from 1000 to 40,000 ppm. In order to extract the sensitivity, we used the change in the sensor voltage level as the output response and the hydrogen concentration as the input measurand. As plotted in Figure 8b, a linear relationship between the response voltage change and the hydrogen concentration was observed from 2000 to 10,000 ppm, beyond which the increase in the response voltage became smaller. The sensitivity was –1.75 µV/ppm, which was derived from the linear regime.

The response and recovery transient characteristics are shown in the magnified plots located on the right-hand side. Definition of response time and recovery time used in this study are illustrated in Figure 9. The response times for both bias conditions were very fast, which were shorter than the sampling period (i.e., 0.4 s). The recovery seemed to be slow, but it should be noted that it takes a long time for the remaining gas inside the gas line to be evacuated completely after the value is closed by a controller. Therefore, the actual recovery times would be much shorter.

In Table 1, the sensor characteristics are compared with other reports made on FET and diode-type hydrogen sensors. Our prototype sensor exhibited excellent sensor response characteristics as well as low standby power consumption.

## 4. Conclusions

A Pd-functionalized hydrogen sensor fabricated on an AlGaN/GaN-on-Si heterostructure platform was investigated under different operation conditions. The optimum reaction temperature was 200 °C. Constant voltage and current mode operations were investigated with different bias conditions. While the constant voltage bias mode operation exhibited more stable operation in the high voltage saturation regime, the constant current bias mode operation exhibited more stable operation in the low current regime. When determining the bias conditions, it is recommended to choose the stable response regime. The constant current bias mode operation in the low current regime is highly recommended considering both stability and standby power consumption. A prototype sensor was fabricated and evaluated in this study and exhibited excellent sensing characteristics with stable operation. The standby power consumption was extremely low (i.e., 0.15 mW/mm) owing to the low standby current and voltage levels. Such excellent characteristics resulted from the stable current bias mode operation in conjunction with the capability for high-temperature operation.

## Figures and Tables

**Figure 1 sensors-19-05549-f001:**
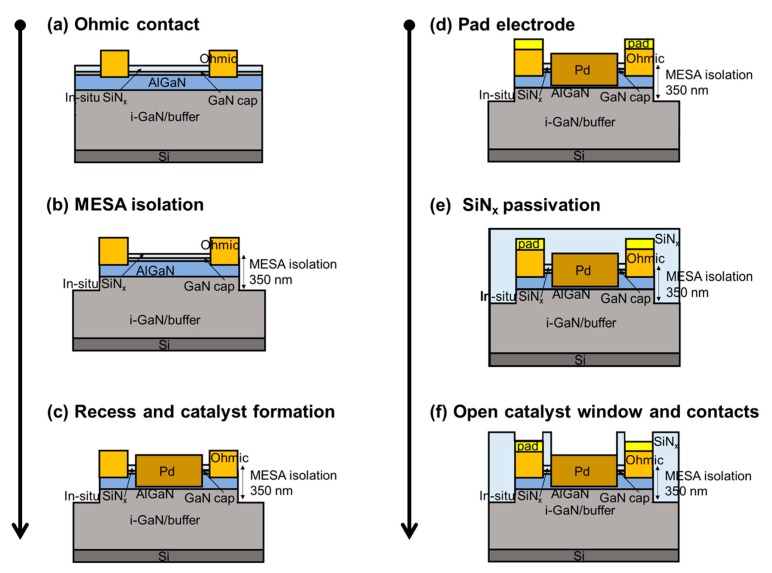
Fabrication process of a Pd-functionalized AlGaN/GaN-on-Si hydrogen sensor: (**a**) Ohmic contact, (**b**) MESA isolation, (**c**) recess and catalyst formation, (**d**) pad electrode, (**e**) SiN_x_ passivation, and (**f**) open catalyst window and contacts.

**Figure 2 sensors-19-05549-f002:**
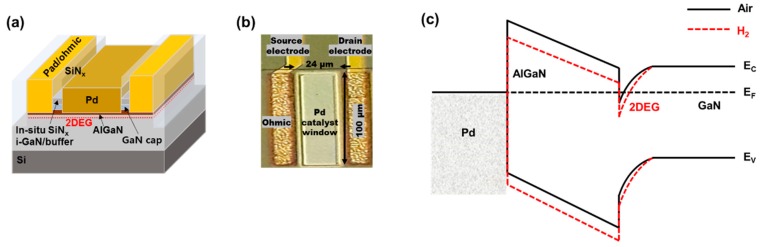
(**a**) 3D schematic of Pd-functionalized AlGaN/GaN-on-Si hydrogen sensor, (**b**) microscopic top view of a fabricated sensor, and (**c**) change of AlGaN/GaN energy band structure with hydrogen absorption.

**Figure 3 sensors-19-05549-f003:**
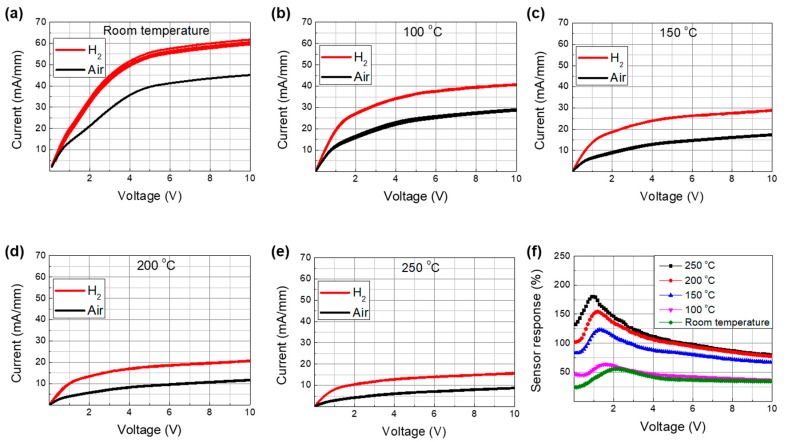
Current–voltage characteristics of the fabricated Pd-functionalized AlGaN/GaN-on-Si hydrogen sensor with and without hydrogen injection at various temperatures: (**a**) Room temperature, (**b**) 100 °C, (**c**) 150 °C, (**d**) 200 °C, and (**e**) 250 °C. (**f**) Response characteristics of the fabricated Pd-functionalized AlGaN/GaN-on-Si hydrogen sensor versus bias voltage as a function of temperature, which were extracted from (**a**) to (**e**).

**Figure 4 sensors-19-05549-f004:**
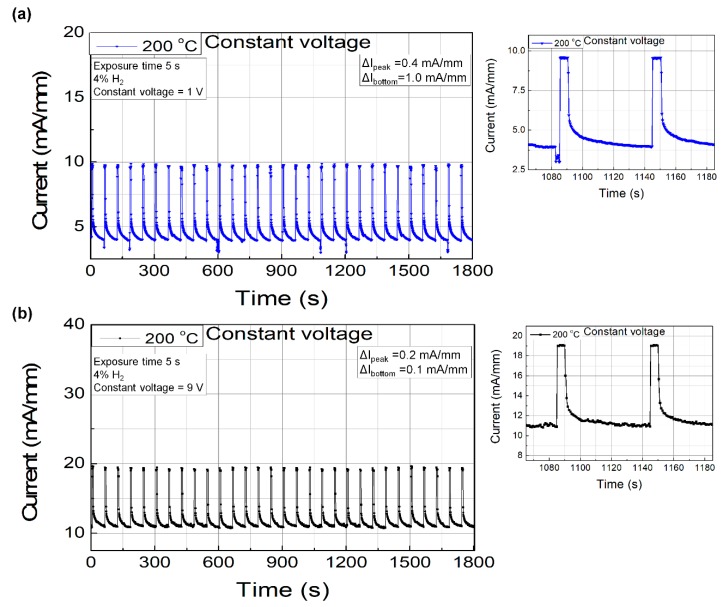
Sensing repeatability characteristics of the fabricated Pd-functionalized AlGaN/GaN-on-Si hydrogen sensor with bias voltages of (**a**) 1 V and (**b**) 9 V. The measurement temperature was 200 °C.

**Figure 5 sensors-19-05549-f005:**
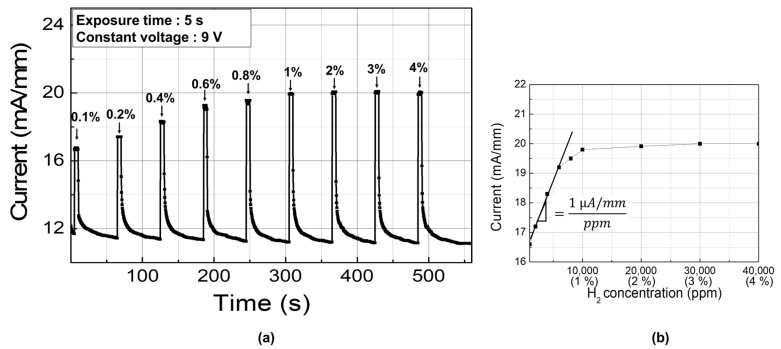
(**a**) Response characteristics as a function of hydrogen concentration and (**b**) their corresponding response plot.

**Figure 6 sensors-19-05549-f006:**
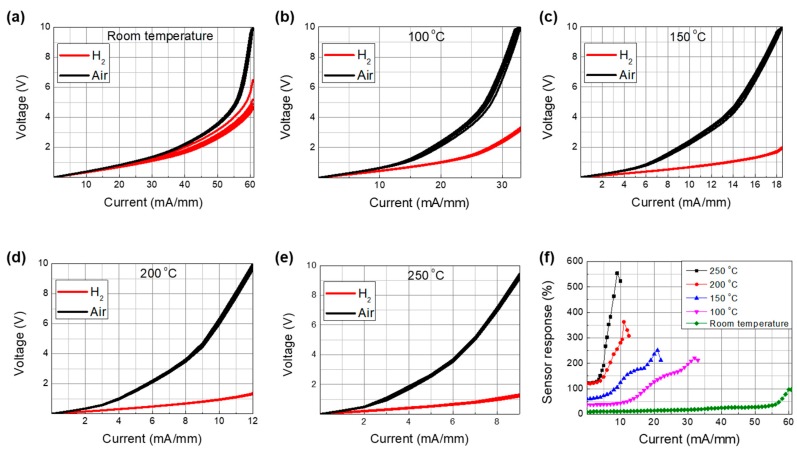
Voltage–current characteristics of the fabricated Pd-functionalized AlGaN/GaN-on-Si hydrogen sensor with and without hydrogen injection at various temperatures: (**a**) Room temperature, (**b**) 100 °C, (**c**) 150 °C, (**d**) 200 °C, and (**e**) 250 °C. (**f**) Response characteristics of the Pd-functionalized AlGaN/GaN-on-Si hydrogen sensor versus bias current as a function of temperature, which were extracted from (**a**) to (**e**).

**Figure 7 sensors-19-05549-f007:**
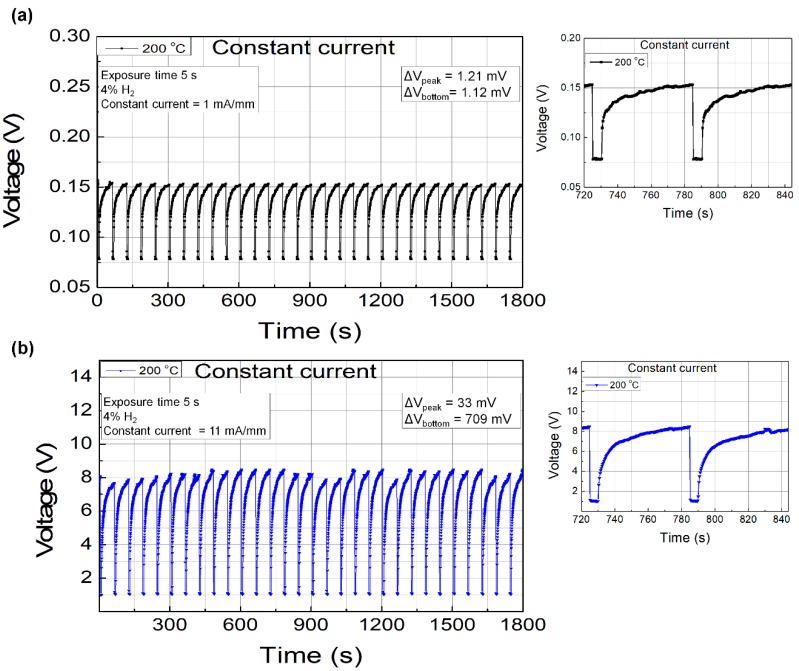
Sensing repeatability characteristics of the fabricated Pd-functionalized AlGaN/GaN-on-Si hydrogen sensor with bias currents of (**a**) 1 mA/mm and (**b**) 11 mA/mm. The measurement temperature was 200 °C.

**Figure 8 sensors-19-05549-f008:**
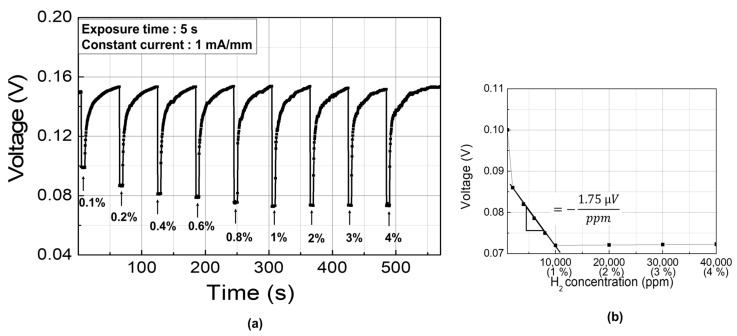
(**a**) Response characteristics as a function of hydrogen concentration and (**b**) their corresponding response plot.

**Figure 9 sensors-19-05549-f009:**
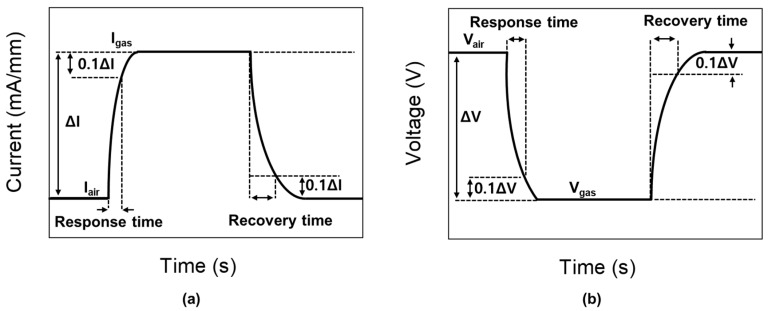
Definition of response time and recovery time for (**a**) constant voltage bias mode and (**b**) constant current bias mode.

**Table 1 sensors-19-05549-t001:** Comparison of sensor characteristics with other reports.

Sensor Platform	Temp.	Hydrogen Concentration	Response Time	Recovery Time	Sensor Response	Power Consumption	Ref.
Diode (AlGaAs)	Room temp.	1%	58 s	-	155.9%	-	[23]
Diode (GaN)	Room temp.	1%	15 s	19 s	1 × 10^5^%	-	[24]
Diode (GaN)	200 ℃	4%	-	-	7 × 10^8^%	-	[19]
Diode (GaN)	300 ℃	0.081%	25.1 s	34.1 s	0.11%	-	[25]
Diode (AlGaN/GaN)	Room temp.	0.05%	-	-	2.4%	-	[26]
Diode (AlGaN/GaN)	Room temp.	4%	-	-	3700%	382 W/cm^2^	[27]
MOS diode (AlGaN/GaN)	Room temp.	10%	~30 s	-	-	5333 W/cm^2^	[28]
FET (Si)	150 ℃	0.5%	18 s	19 s	-	35.8 mW (Sensor area N/A)	[29]
FET (AlGaN/GaN)	200 ℃	4%	3 s	-	72%	3.93 W/cm^2^	[30]
FET (AlGaN/GaN) Constant voltage	200 ℃	4%	< 0.4 s	12.4 s	80%	347 W/cm^2^	This work
FET (AlGaN/GaN) Constant current	200 ℃	4%	< 0.4 s	27.2 s	120%	0.54 W/cm^2^	This work

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
