# Peer review of "Investigation of Stability and Power Consumption of an AlGaN/GaN Heterostructure Hydrogen Gas Sensor Using Different Bias Conditions"

_sensors, 2019, doi:10.3390/s19245549_

Round 1

Reviewer 1 Report

This work reported a Pd-functionalized hydrogen sensor fabricated on an AlGaN/Gan-on-Si heterostructure platform. The authors performed systematic studies to understand the mechanism. Overall, it is an interesting research topic. However, the following questions need to be clarified before the work can be considered for publication:

Have the authors studied the selectivity of the sensing platform? How does the result compare to the state-of-the-art hydrogen sensors reported previously? The authors need to provide detailed explanations on why the sensitivity can change by varying sensing conditions. For example, why 200 °C the optimum sensing temperature?

Reviewer 2 Report

The manuscript reports the a Pd-functionalized hydrogen sensor fabricated on an AlGaN/GaN-on-Si heterostructure platform. A prototype sensors showed excellent sensing characteristics with stable operation with low standby power consumption (0.15 mW/mm) owing to low levels of standby current and voltage

I recommend the publication of this manuscript if the following issues are clearly addressed.

AlGaN/GaN-based various gas sensor research has been reported a lot. Is using Pd the novelty of this study? It is necessary to describe in the introduction the differences from other studies. On page 2, line 53, a rapid thermal annealing is described at 830 ° It is necessary to improve the quality of the paper by describing the necessity of this process. On page 4, line 127, the response times are described as 0.4 s, which seems to describe the rise time. The definition of rise time varies from article to article. Therefore, it is necessary to mention in the manuscript how rise time is defined. In addition to the rise time, the decay time needs to be calculated. The authors mentioned that the power consumption value of 0.15 mW /mm is extremely low without any references. Therefore, comparison with other papers is necessary to prove this point.

Reviewer 3 Report

The paper reports on the realization of a AlGaN/GaN-on-Si 9 heterostructure platform for H2 chemical sensing. Sensor characterizations are performed using two different bias modes: constant voltage bias mode and constant current bias mode.

The topic is appropriate for publication in the journal, however I have found different points that weaken the technical and scientific soundness of the paper.

First of all the “sensitivity” definition. Actually, what the authors define sensitivity is the sensor response of a chemical sensor.  In the sensor community, one can obtain the sensitivity curve by a derivative procedure from the response curve (if it is linear). Sensor Response is defined as the output response of the sensor as a function of the measurand applied to its input. The following references could be useful to investigate these aspects in detail:  

D’Amico, C. Di Natale, A contribution on some basic definitions of sensors properties, IEEE Sens. J. (2001) 183-190.

Sharma, M. Tomar and V. Gupta, Journal of Materials Chemistry, 2012, 22, 23608-23616.

In this sense, a sentence like “maximum sensitivity of 180% was achieved at 250” has no scientific sense.

Once stated the definition of sensor response, a chemical sensing characterization requires also a sensorgram, a dynamic measurement of the sensor response, in order to be aware of the dynamics of the process and, eventually, reversibility. Here it is difficult to recognize a sensor response: maybe figure 4? If yes, it is not clear the single sensing cycle and why saturation is not reached. Why sensor response is calculated after 5s? A deep discussion on the sensing mechanism as a comment of figure 2 is highly required. Moreover, figures 4f and 5f have been recorded at saturation? It is really unclear.

Moreover, it is not clear the motivation behind the two different bias modes.

Finally there is no comparison with other literature results.

For all the above observations I do not think the paper can be accepted for publication in the present form. I would suggest the authors an hard revision of the paper and a re-submission.

Round 2

Reviewer 2 Report

The manuscript by Choi et al. titled ‘Investigation of Stability and Power Consumption of AlGaN GaN Heterostructure Hydrogen Gas Sensor Using Different Bias Conditions’ has been thoroughly revised and improved. The contents in the manuscript will provide valuable information for inorganic electronics community. I recommend to accept this revised manuscript for publication in Sensors.

Reviewer 3 Report

I am satisfied of the revision process made by the authors.

As for me the paper can be accepted for publication in the journal.